# Chemotherapy and Heart-Specific Mortality in Elderly Men with Prostate Cancer: A Propensity Score Matching Analysis

**Chenghao Zhanghuang**[1,2,3,4,5†], **Huake Wang**[1,2†], **Jinkui Wang**[3,4†], **Li Li**[6], **Jinrong Li**[1,2], **Zipeng Hao**[1,2], **Jiacheng Zhang**[1,2], **Ling Liu**[7*], **Bing Yan**[1,2*]

1 Department of Urology, Kunming Children's Hospital (Children's Hospital affiliated to Kunming Medical University), Kunming, PR China, 2 Yunnan Province Clinical Research Center for Children's Health and Disease, Kunming, PR China, 3 Department of Urology, Children's Hospital of Chongqing Medical University, Chongqing, PR China, 4 Children's Hospital of Chongqing Medical University, Chongqing, PR China, 5 Yunnan Key Laboratory of Children's Major Disease Research, Kunming Children's Hospital (Children's Hospital affiliated to Kunming Medical University), Kunming, PR China, 6 Department of Science and Education, Kunming Children's Hospital (Children's Hospital affiliated to Kunming Medical University), Kunming, PR China, 7 Department of Neonatal, Kunming Children's Hospital (Children's Hospital affiliated to Kunming Medical University), Kunming, PR China

† These authors have contributed equally to this work and should be considered co-first authors.
* ynll2012@163.com (LL); ybwcy@163.com (BY)

## Abstract

### Objective

Prostate cancer (PC) is the most common malignant tumour in men, and atherosclerotic cardiovascular disease (ASCVD) is the leading cause of non-cancer death in PC patients. The main purpose of this study was to investigate whether chemotherapy increases heart-specific mortality (HSM) in elderly patients with PC.

### Methods

Patient information was downloaded from the Surveillance, Epidemiology, and End Results (SEER) database from 2010 to 2018. We included all elderly patients with PC. The multivariate logistic regression model was used to explore the influencing factors of patients receiving chemotherapy. Confounders were excluded using a 1:1 proportional propensity score match, and a competing risk model and cumulative incidence plot were used to analyze HSM and other cause mortality (OCM) in patients who received chemotherapy versus those who did not.

### Results

A total of 135183 elderly prostate patients were enrolled in this study, of whom 1361 received chemotherapy. The multivariate logistic regression model showed that older patients were more likely to not receive chemotherapy, married patients were more likely to receive chemotherapy, and the higher the TNM stage and tumor histological grade, the more patients received chemotherapy. In the original cohort before unmatched, there was no significant difference in HSM between chemotherapy and non-chemotherapy patients

**Data availability statement:** All patient information files are available from the Surveillance, Epidemiology, and End Results (SEER) database (URLs: https://seer.Cancer.gov/).

**Funding:** This study was supported by Yunnan Education Department of Science Research Fund (No. 2020J0228, 2023J0295), Kunming Medical Joint Project of Yunnan Science and Technology Department (No. 202001AY070001-271, 202301AY070001-108), Famous Doctors of Yunnan Province 'Xingdian Talents Support Program' (No. XDYC-MY-2022-0096), Open Research Fund of Clinical Research Center for Children's Health and Diseases of Yunnan Province (2022-ETYY-YJ-03), and Science and Technology project of Kunming Municipal Commission of Health and Construction (No.2024-SW (lead) -7). The funding bodies played no role in the study's design and collection, analysis and interpretation of data, and writing the manuscript.

**Competing interests:** The authors have declared that no competing interests exist.

(P = 0.27). After 1:1 matching, HSM was significantly higher in patients without chemotherapy than in patients with chemotherapy (HR 2.54; P =0.002).

## Conclusions

Our results indicate that HSM is significantly higher in patients without chemotherapy than in those with chemotherapy. Therefore, although chemotherapy can lead to cardiotoxicity in elderly patients with PC, chemotherapy does not increase the HSM of patients and will benefit patients in the long-term survival.

## Background

Prostate cancer (PCa) is the most frequently diagnosed malignant neoplasm among men, and after two decades of decline, its incidence rate has been increasing by 3% annually from 2014 to 2019 [1]. According to the statistical data from the American Cancer Society in 2023, prostate cancer ranks second in incidence among American men, just after skin cancer, and is the third leading cause of cancer-related mortality in males. In 2023, the United States is projected to have over 288,300 new cases and 34,700 death cases of prostate cancer [1]. As we progress into 2024, the global burden of prostate cancer continues to rise. It is estimated that in 2024, the United States will see 299,010 new cases and 35,250 death cases of prostate cancer [2]. The number is expected to increase to 5.02 million by 2030 [3]. By 2040, the number of cancer patients in the world will exceed 28.4 million, an increase of 47% compared with 2020, which will bring great obstacles to world economic development and human life [4]. It is worth noting that with the popularization of early screening, the diagnosis rate of most cancers is greatly improved, and the diseases are mainly in the early stage. Urine biopsy is an attractive and promising method for the detection of PC. In addition to urine, biomarkers including androgen receptor variants, bone metabolism, neuroendocrine and metabolites in serum of PC patients also have the role of early diagnosis of PC [5]. However, among newly diagnosed patients with prostate cancer and breast cancer, there are still significant parts in the advanced stage of the disease [1]. A large number of cases, and the second-highest cancer death rate, have led clinicians to pay more attention to PC treatment.

Similar to most solid tumour, elderly patients suffer from higher incidence and mortality of prostate cancer, with the median age of diagnosis being 66 years old [6]. In addition, with the aggravation of the aging population, the base of the elderly population is also expanding, and the health management of cancer in the elderly population has become a major problem that cannot be ignored [7]. But the difference from most solid tumour, is that chemotherapy is not the primary treatment for PC. Instead, androgen-reducing therapies have been in place for more than a century [8]. Androgen deprivation therapy (ADT) has been used for more than a century [9]. In addition, abiraterone and prednisolone combined with ADT can be used as a new standard treatment for high-risk non-metastatic PC patients. The results showed a significantly higher rate of metastasis-free survival compared with ADT alone [10]. However, these treatments are increasingly found to be ineffective in controlling the disease. For nearly a decade, chemotherapy in the treatment of prostate cancer play a key role in more and more, from a palliative treatment to hormone sensitivity important auxiliary treatment for prostate cancer, can reduce the pain of patients caused by cancer, and significantly improve the patient's survival, became the centre of advanced metastatic prostate cancer treatment method [11]. More importantly, indications for chemotherapy have expanded again since 2014. Studies have shown that chemotherapy can also benefit patients with early-stage PC

[12]. However, it is well known that chemotherapy drugs become a "double-edged sword" in the treatment of cancer patients due to their inevitable drug resistance and cardiotoxicity [13].

Cancer-specific mortality (CSM) in patients with PC has declined as treatment modalities have improved. However, non-cancer deaths remained stable. Among them, heart-specific mortality (HSM) has gradually become an important cause of non-cancer death in cancer patients [14]. Atherosclerotic cardiovascular disease (ASCVD) is the leading non-cancer cause of death in PC patients. Data show that two-thirds of PC patients have a high cardiovascular risk, and almost a quarter of them have been associated with cardiovascular injury to some degree at the time of diagnosis, and the proportion is even higher in the elderly population [15]. Unfortunately, to our knowledge, the advantages and disadvantages of chemotherapeutic agents in elderly patients with PC have not been reported.

Based on the above situation, we collected the 2010–2018 of elderly PC patients from Surveillance, Epidemiology, and End Results (SEER) database, conducted propensity matching and evaluated the advantages and disadvantages of chemotherapy, and provided suggestions for reducing related mortality in clinical diagnosis and treatment.

## Methods

### Data source and study population

Patient data for PC were downloaded from the SEER database. The SEER database is the U.S. national cancer database that includes 18 cancer registries covering approximately 30% of the U.S. population. SEER database data is public and patient information is anonymous, so our study does not require ethical approval or informed consent of patients. Our research follows the regulations published by the SEER database.

We collected clinical data and follow-up information from patients over 65 years of age with prostate cancer from 2010 to 2018. Inclusion criteria:(1) pathological diagnosis of prostate cancer; (2) Aged 65 or above. Exclusion criteria:(1) TNM staging is unknown; (2) Unknown surgical method; (3) Tumor grade is unknown; (4) Prostate specific antigen (PSA) and Gleason scores are unknown; (5) Survival time less than 1 month. The screening flow chart of patients is shown in Fig 1.

### Study variables

Clinicopathological information and follow-up information were collected. We collected demographic information on the patients, including age, race, and marital status. Patient tumour information included tumour histological grade, TNM stage, PSA level, and Gleason score. The patient's treatment information included surgical procedures, chemotherapy, and radiation. Follow-up information included time of survival, cause of death, and survival status.

Patients were categorized by race as white, black, and other (American Indian/AK Indian, Asian/Pacific Islander). The marital status of the patients was classified as married or unmarried. Histological grades include highly differentiated (grade I), moderately differentiated (grade II), poorly differentiated (grade III), and undifferentiated (grade IV). The surgical classification includes nonoperative, partial excision, and radical excision. Causes of death included heart-specific mortality (HSM) and other causes of mortality (OCM). PSA is classified as low (< 4ng/ mL), medium (4–10ng/ml), and high (> 10ng/ml). Gleason scores were classified as ≤ 6,4 + 3,3 +4, and ≥8.

### Statistical analyses

The continuity variable (age) was described by means and standard deviations, and the t-test was used for comparison between groups. Classification variables were described by

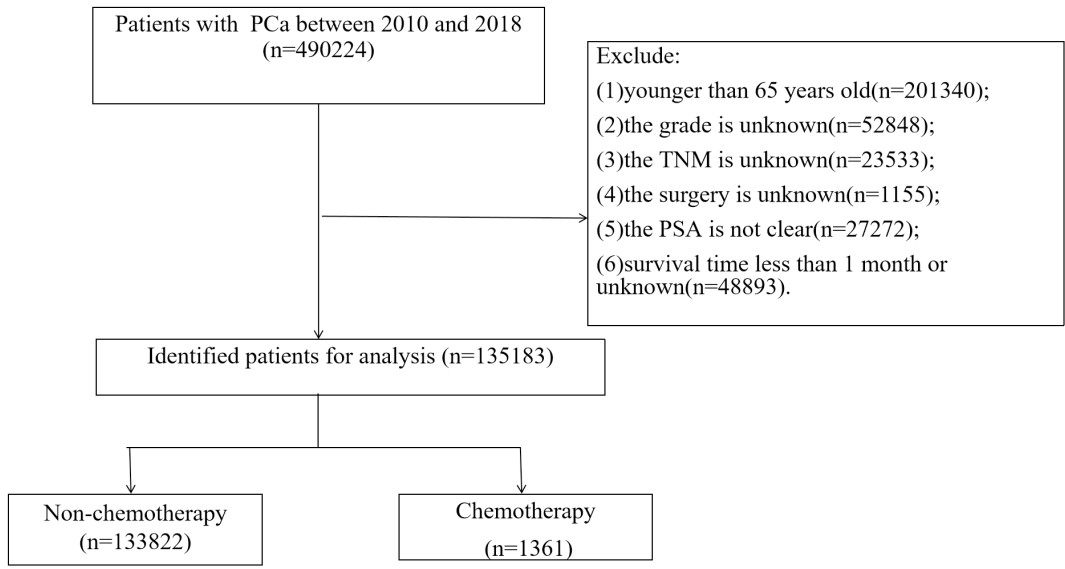

**Fig 1. Consolidated Standards of Reporting Trials flowchart of study inclusion criteria.**

frequency, and the chi-square test was used for comparison between groups. In a cohort of all patients, we used a multivariate logistic regression model to analyze factors influencing chemotherapy and non-chemotherapy. Variables included age (continuous), race (white vs black vs other), marital status (married vs unmarried), T staging (T1 vs T2 vs T3 vs T4), N staging (N0 vs N1), M stage (M0 vs M1), histological grade (I vs II vs III vs IV), PSA (<4ng/ml vs 4–10 ng/ml vs > 10ng/ mL), Gleason score (≤6 vs 4+3 vs 3+4 vs ≥8).

First, depending on whether the patient is receiving chemotherapy, we use nearest neighbour 1:1 matching to obtain a 1:1 cohort, which will be more reliable. Propensity score matching was performed for age, race, tumour histological grade, TNM stage, surgery, radiotherapy, PSA, and Gleason score to balance these confounding factors. We then used cumulative incidence plots and multivariable competitive risk regression (CRR) models to investigate the effects of chemotherapy and no chemotherapy on HSM and OCM in patients. The competitive risk model refers to a situation in which the occurrence of other events prevents us from observing the occurrence of the event, thus changing the probability of the observed event. We used a competitive risk model to explore the independent risk factors for HSM and OCM in elderly prostate cancer patients. We also analyzed whether chemotherapy affected HSM in patients with 1:1 propensity score matching.

Subsequently, sensitivity analysis was used to optimize the matching results to increase the reliability of our results. (1) In the unmatched cohort, the CRR model was used to compare HSM differences between chemotherapy and non-chemotherapy patients; (2) HSM difference between chemotherapy and non-chemotherapy patients was compared under 1:2 propensity score matching; (3) The difference in HSM between patients with and without chemotherapy was compared under the propensity score matching of 1:4; (4) HSM differences between chemotherapy and non-chemotherapy patients were compared using inverse probability weighted propensity score matching.

All statistical analyses were conducted using SPSS software (version 23.0, SPSS, Chicago, IL, USA) and R version 3.4.1. Propensity score matching is carried out by R package "non-random", "reshape2", "Matching" and "survey". The competitive risk model was based on

"CMPRSK", "RMS" and "SURVIVAL". A P value less than 0.05 was considered statistically significant.

## Results

### Clinical features

A total of 135183 elderly prostate patients were included in the study, including 1361 who received chemotherapy and 133822 who did not. The basic information and clinicopathological features of all patients were shown in Table 1. It can be found that the higher the TNM stage and the tumor histological grade, the higher the number of patients receiving chemotherapy. PSA levels were higher in patients receiving chemotherapy. In addition, the group of patients who received chemotherapy had a higher proportion of patients who did not receive radiotherapy and did not undergo surgery.

### Multivariate logistic regression analysis of the original study cohort

In The Unmatched Original Cohort, We Used a multivariate logistic regression model to analyze the factors affecting patients' chemotherapy. We found that older patients were more likely to avoid chemotherapy (OR 0.927, 95%CI 0.917–0.936; P< 0.001), married patients were more likely to receive chemotherapy (OR 1.315, 95%CI 1.159–1.492; P< 0.001), the higher the TNM stage and tumor histological grade, the more patients received chemotherapy. Multivariate logistic regression analysis results are shown in Table 2.

### Multivariate competitive risk regression model after propensity score matching

In the original cohort before unmatched, there was no significant difference in HSM between chemotherapy and non-chemotherapy patients (P = 0.27), while OCM in non-chemotherapy patients was significantly higher than that in chemotherapy patients (P< 0.001) (Fig 2). Then, using nearest neighbour 1:1 matching, we obtained 1361 patients who received chemotherapy and 1361 patients who did not receive chemotherapy for the subsequent study. The matching results are shown in Table 1, and the propensity density before and after matching is shown in Fig 3, indicating that the basic characteristics of patients after matching are almost the same. A multivariate CRR model was used to analyze HSM and OCM in the 1:1 matched cohort of chemotherapy and non-chemotherapy patients, and the results showed that HSM in patients without chemotherapy was significantly higher than that in patients with chemotherapy (HR 0.394 P =0.002)(Fig 4). The results of the multivariate CRR model analysis are shown in Table 3, showing the independent risk factors of HSM and OCM respectively.

### Sensitivity analyses

Subsequently, we optimized propensity matching and used multivariable CRR to compare HSM in patients who received chemotherapy and those who did not. (1) In the original cohort, there was no significant difference in HSM between patients who received chemotherapy and those who did not (HR 1.63, P=0.27); (2) In the 1:2 cohort, the HSM of patients who did not receive chemotherapy was significantly higher than that of patients who received chemotherapy (HR 0.468, P = 0.006); (3) In the 1:4 cohort, HSM of patients who did not receive chemotherapy was also significantly higher than that of patients who received chemotherapy (HR 0.519, P = 0.013). (4) We used the inverse probability propensity weighted matching method to balance the influence of confounding factors, and the results showed that the HSM

**Table 1. Sociodemographic and clinical characteristics of patients receiving chemotherapy versus Non-chemotherapy in PC patients.**

| | Unmatched | | | | Matched | | | |
|---|---|---|---|---|---|---|---|---|
| | Non-chemotherapy N=133822 | Chemotherapy N=1361 | p | SMD | Non-chemotherapy N=1361 | Chemotherapy N=1361 | p | SMD |
| Age (mean (SD)) | 71.58 (5.51) | 71.62 (5.43) | 0.791 | 0.007 | 72.08 (5.83) | 71.62 (5.43) | 0.03 | 0.083 |
| Race (%) | | | | | | | | |
| white | 106269 (79.4) | 1112 (81.7) | 0.015 | 0.084 | 1121 (82.4) | 1112 (81.7) | 0.628 | 0.037 |
| black | 15845 (11.8) | 160 (11.8) | | | 163 (12.0) | 160 (11.8) | | |
| other | 11708 (8.7) | 89 (6.5) | | | 77 (5.7) | 89 (6.5) | | |
| Marital (%) | | | | | | | | |
| No | 43171 (32.3) | 422 (31.0) | 0.34 | 0.027 | 428 (31.4) | 422 (31.0) | 0.836 | 0.01 |
| Married | 90651 (67.7) | 939 (69.0) | | | 933 (68.6) | 939 (69.0) | | |
| Grade (%) | | | | | | | | |
| I | 19984 (14.9) | 37 (2.7) | <0.001 | 1.001 | 33 (2.4) | 37 (2.7) | 0.729 | 0.044 |
| II | 54285 (40.6) | 141 (10.4) | | | 134 (9.8) | 141 (10.4) | | |
| III | 57644 (43.1) | 1147 (84.3) | | | 1165 (85.6) | 1147 (84.3) | | |
| IV | 1909 (1.4) | 36 (2.6) | | | 29 (2.1) | 36 (2.6) | | |
| T (%) | | | | | | | | |
| T1 | 61367 (45.9) | 452 (33.2) | <0.001 | 0.581 | 445 (32.7) | 452 (33.2) | 0.466 | 0.061 |
| T2 | 52455 (39.2) | 423 (31.1) | | | 434 (31.9) | 423 (31.1) | | |
| T3 | 17865 (13.3) | 277 (20.4) | | | 298 (21.9) | 277 (20.4) | | |
| T4 | 2135 (1.6) | 209 (15.4) | | | 184 (13.5) | 209 (15.4) | | |
| N (%) | | | | | | | | |
| N0 | 128631 (96.1) | 829 (60.9) | <0.001 | 0.949 | 843 (61.9) | 829 (60.9) | 0.609 | 0.021 |
| N1 | 5191 (3.9) | 532 (39.1) | | | 518 (38.1) | 532 (39.1) | | |
| M (%) | | | | | | | | |
| M0 | 128597 (96.1) | 449 (33.0) | <0.001 | 1.755 | 434 (31.9) | 449 (33.0) | 0.567 | 0.024 |
| M1 | 5225 (3.9) | 912 (67.0) | | | 927 (68.1) | 912 (67.0) | | |
| Surgery (%) | | | | | | | | |
| No | 86748 (64.8) | 1095 (80.5) | <0.001 | 0.492 | 1107 (81.3) | 1095 (80.5) | 0.089 | 0.084 |
| Local tumor excision | 7894 (5.9) | 125 (9.2) | | | 96 (7.1) | 125 (9.2) | | |
| Radical prostatectomy | 39180 (29.3) | 141 (10.4) | | | 158 (11.6) | 141 (10.4) | | |
| Radiation (%) | | | | | | | | |
| No | 80459 (60.1) | 948 (69.7) | <0.001 | 0.201 | 978 (71.9) | 948 (69.7) | 0.222 | 0.048 |
| Yes | 53363 (39.9) | 413 (30.3) | | | 383 (28.1) | 413 (30.3) | | |
| PSA (%) | | | | | | | | |
| <4 | 13061 (9.8) | 108 (7.9) | <0.001 | 0.94 | 64 (4.7) | 108 (7.9) | 0.001 | 0.147 |
| 4-10 | 76891 (57.5) | 249 (18.3) | | | 226 (16.6) | 249 (18.3) | | |
| >10 | 43870 (32.8) | 1004 (73.8) | | | 1071 (78.7) | 1004 (73.8) | | |
| Gleason (%) | | | | | | | | |
| ≤6 | 5131 (3.8) | 10 (0.7) | <0.001 | 0.626 | 10 (0.7) | 10 (0.7) | 0.618 | 0.062 |
| 3+4 | 17164 (12.8) | 18 (1.3) | | | 13 (1.0) | 18 (1.3) | | |
| 4+3 | 9523 (7.1) | 19 (1.4) | | | 20 (1.5) | 19 (1.4) | | |
| ≥8 | 7143 (5.3) | 74 (5.4) | | | 91 (6.7) | 74 (5.4) | | |
| Unknown | 94861 (70.9) | 1240 (91.1) | | | 1227 (90.2) | 1240 (91.1) | | |

**Table 2.  Multivariate Logistic regression model to predict chemotherapy or non-chemotherapy in PC patients.**

|  | OR | 95% CI |  | P |
|---|---|---|---|---|
| Age | 0.927 | 0.917 | 0.936 | <0.001 |
| Race |  |  |  |  |
| white | Reference |  |  |  |
| black | 0.748 | 0.624 | 0.897 | 0.002 |
| other | 0.701 | 0.556 | 0.883 | 0.003 |
| Marital |  |  |  |  |
| No | Reference |  |  |  |
| Married | 1.315 | 1.159 | 1.492 | <0.001 |
| Grade |  |  |  |  |
| I | Reference |  |  | <0.001 |
| II | 1.472 | 1.02 | 2.126 | 0.039 |
| III | 3.896 | 2.749 | 5.522 | <0.001 |
| IV | 4.695 | 2.819 | 7.819 | <0.001 |
| T |  |  |  | <0.001 |
| T1 | Reference |  |  |  |
| T2 | 0.984 | 0.851 | 1.139 | 0.832 |
| T3 | 1.382 | 1.15 | 1.66 | 0.001 |
| T4 | 1.649 | 1.348 | 2.017 | <0.001 |
| N |  |  |  |  |
| N0 | Reference |  |  |  |
| N1 | 2.286 | 1.992 | 2.624 | <0.001 |
| M |  |  |  |  |
| M0 | Reference |  |  |  |
| M1 | 19.055 | 16.323 | 22.245 | <0.001 |
| PSA |  |  |  | <0.001 |
| <4 | Reference |  |  |  |
| 4-10 | 0.446 | 0.352 | 0.567 | <0.001 |
| >10 | 0.733 | 0.586 | 0.916 | 0.006 |
| Gleason |  |  |  | <0.001 |
| ≤6 | Reference |  |  |  |
| 3+4 | 0.324 | 0.149 | 0.708 | 0.005 |
| 4+3 | 0.411 | 0.188 | 0.899 | 0.026 |
| ≥8 | 1.095 | 0.55 | 2.182 | 0.796 |
| Unknown | 1.776 | 0.935 | 3.375 | 0.079 |

was significantly higher in patients who did not receive chemotherapy than in those who received chemotherapy (P=0.0028) (Fig 5).

## Discussion

In recent years, long-term survival rates for a range of cancers, such as PC, have improved as the population ages, life expectancy increases and early cancer detection rates rise. Correspondingly, the incidence of cardiotoxic-induced death in the course of treatment also showed an increasing trend [16]. Overall, tumour treatment-related cardiovascular toxicity in clinical studies can lead to cardiac insufficiency in up to 26% of subjects [17]. With the prolongation of treatment duration and survival, cardiotoxicity caused by tumour treatment has become the second leading cause of death for tumour survivors in addition to recurrence and metastasis

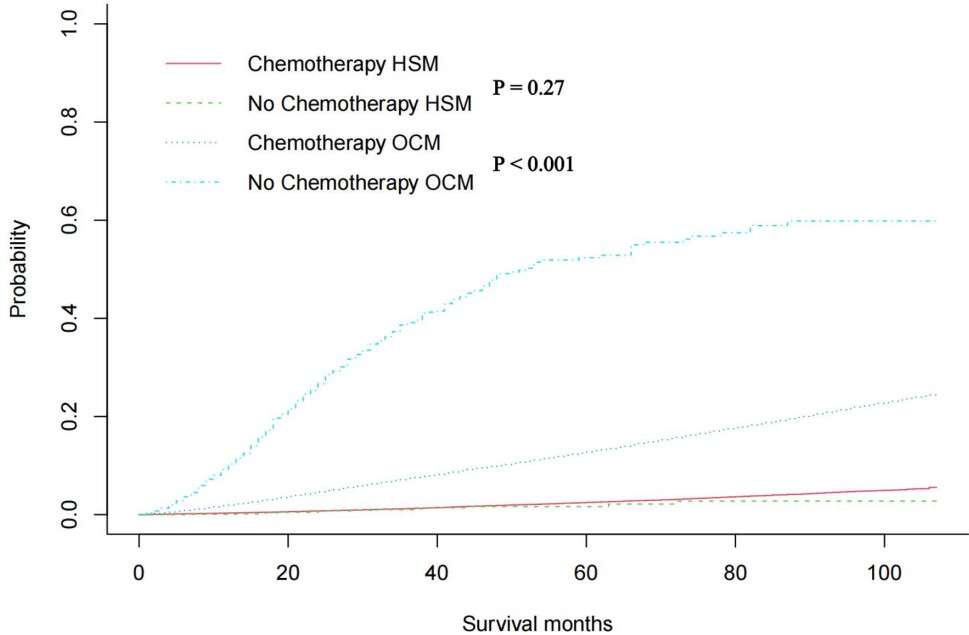

**Fig 2. Cumulative incidence plots in the original cohort showed HSM and OCM rates in PC patients.**

[17]. "Cardio-Oncology" is a new discipline, and HSM caused by cardiotoxicity related to tumour therapy is gradually attracting attention.

Previous studies have found that PC has a higher incidence and mortality rate in the elderly. With the significant extension of the average life span of the world's population, the proportion of the elderly is also increasing in the total population. Since age is the biggest risk factor for prostate cancer, the disease is understandably one of the biggest public health problems [18]. While the average life expectancy for men is 70–75 years, those who reach this average can expect to live another 14 years, those aged 80–85 can expect to live another 8 years, and those aged 85 can expect to live another 6 years [19]. The treatment of prostate cancer patients is based on patients with a life expectancy of more than 10 years. Therefore, the change in the understanding of the life expectancy of elderly patients leads to the need to update the treatment of elderly patients in this group. There is no consensus on defining the age of elderly patients, but more than 60% of initial cancer diagnoses and more than 70% of cancer deaths occur in patients over 65 years old [20]. To improve the representativeness of our study, PC patients over 65 years old were included in this study.

To improve the prognosis of prostate cancer in older men, many clinical trials have tested the efficacy of adding chemotherapy to active surveillance, local treatment, radical prostatectomy (RP), or radiotherapy (RT). Happily, most attempts have yielded good results. Cooper et al. found that the 5-year biochemical recurrence rate of PCa in the high-risk group was 32%-70%, and proved the safety and effectiveness of chemotherapy for high-risk prostate cancer [21]. Forastiere et al., through a 10-year follow-up study, found that conventional radiotherapy combined with chemotherapy could benefit the survival of locally advanced patients [22]. Tannock et al. found in a prospective study of 1006 patients with metastatic prostate cancer that the chemotherapy drug docetaxel extended the overall survival (OS) of PC patients by three months and reduced PSA by more than 50% in 45% of patients [23]. Rosenthal et al. found in a randomized Phase III clinical trial that compared with traditional radiotherapy combination ADT, the OS of patients with locally advanced prostate cancer could be increased

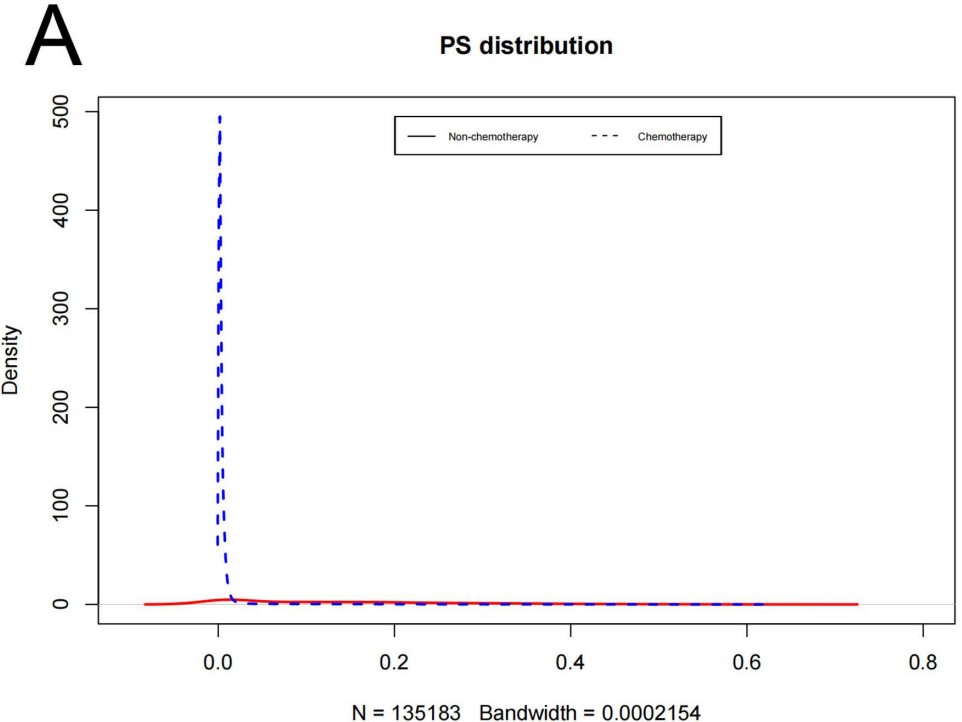

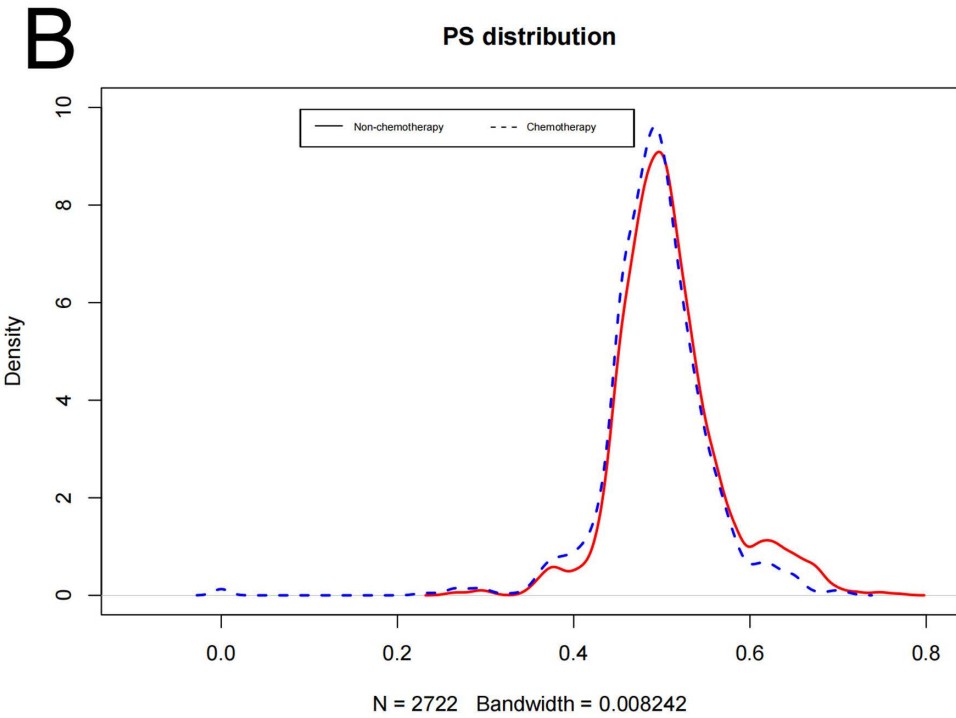

**Fig 3. Density plots of propensity score before (A) and after matching (B).**

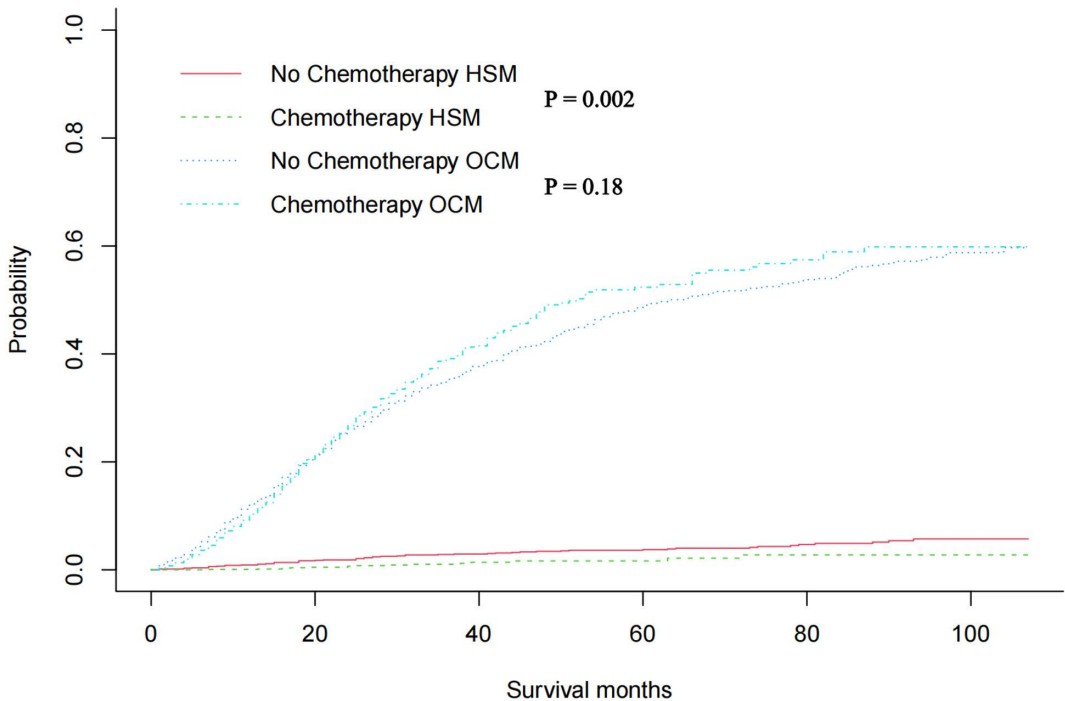

**Fig 4. After 1:1 propensity score matching, Cumulative incidence plots showed HSM and OCM rates in PC patients with or without chemotherapy.**

**Table 3. Propensity score-adjusted competing risk model predict Heart-specific mortality and other causes mortality in PC patients.**

| | Heart-Specific Mortality | | | Other Cause Mortality | | |
|---|---|---|---|---|---|---|
| | HR | 95% CI | p | HR | 95% CI | p |
| Age | 1.049 | 0.223–0.696 | 0.016 | 1.032 | 1.021–1.04 | <0.001 |
| Race | 0.979 | 0.644–1.489 | 0.92 | 0.924 | 0.830–1.03 | 0.14 |
| Marital | 0.611 | 0.377–0.990 | 0.046 | 0.909 | 0.803–1.03 | 0.13 |
| Grade | 1.548 | 0.939–2.552 | 0.087 | 1.147 | 0.933–1.41 | 0.19 |
| T | 1.134 | 0.882–1.459 | 0.33 | 1.144 | 1.078–1.21 | <0.001 |
| N | 0.464 | 0.262–0.822 | 0.008 | 1.137 | 1.005–1.29 | 0.04 |
| M | 1.055 | 0.611–1.821 | 0.85 | 3.736 | 3.085–4.52 | <0.001 |
| Surgery | 0.621 | 0.319 - 1.207 | 0.16 | 1.295 | 1.101–1.52 | 0.001 |
| Radiation | 1.022 | 0.601–1.738 | 0.94 | 1.039 | 0.903–1.20 | 0.59 |
| Chemotherapy | 0.394 | 0.223–0.696 | 0.001 | 1.135 | 1.006–1.28 | 0.04 |
| PSA | 1.174 | 0.818–1.684 | 0.38 | 0.937 | 0.830–1.06 | 0.001 |
| Gleason | 0.836 | 0.362–1.929 | 0.67 | 1.957 | 1.304–2.94 | 0.001 |

from 89% to 93% with chemotherapy [24]. A meta-analysis by Vale et al. showed that the chemotherapy drug docetaxel improved survival in patients with metastatic prostate cancer (mPC) [25].

More importantly, in recent years, stereotactic ablative radiotherapy (SABR) has been recognized as a very promising method for the treatment of prostate cancer patients. The results of a single-center study involving 500 patients with PC showed that SABR was an effective and well-tolerated modality treatment for low - and intermediate-risk PC patients [26]. To

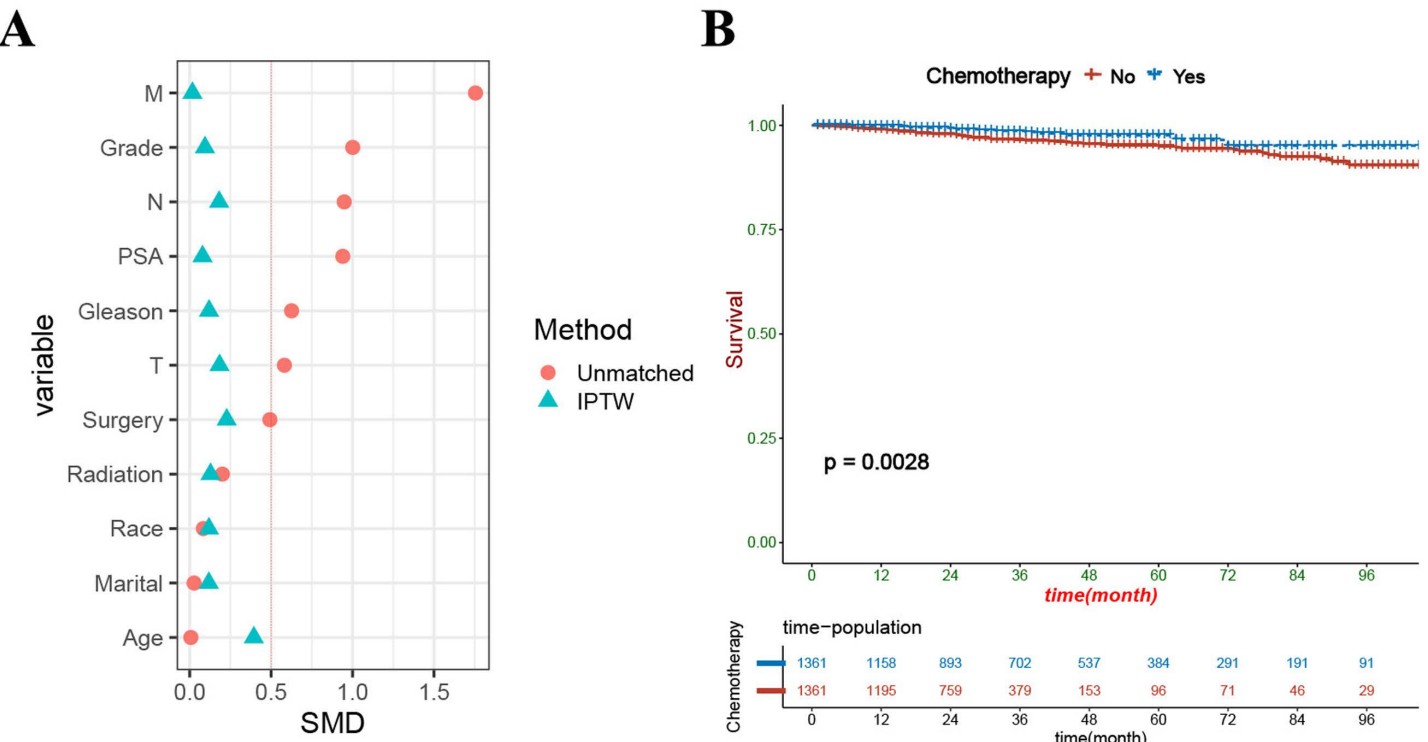

**Fig 5. (A) Standardized mean difference distribution of patients after the inverse probability of treatment weighting matching. (B)**Kaplan-meier curve of PC patients with or without chemotherapy.

further elucidate and standardize the clinical role of SABR in oligometastatic prostate cancer (OMPC), the Italian Society for Radiotherapy and Clinical Oncology (AIRO) established an expert panel to review the existing literature and form a formal consensus. AIRO believes that SABR is safe and effective in the treatment of OMPC [27]. Meanwhile, studies by Mikropoulos et al. have also shown that SABR is a promising treatment for OMPC that has demonstrated clinical benefit in certain clinical Settings, and its use will be expanded in the future [28].

However, some reports have produced the opposite result. Zhao et al. found that neoadjuvant chemotherapy was not a predictor of long-term survival of PC patients, and the use of chemotherapy did not significantly improve prognosis [29]. In a phase II clinical trial of etoposide for locally advanced prostate cancer, Clark et al. [30] found that the anti-tumour function lacking in chemotherapy was significantly improved with a high incidence of thromboembolic adverse events (17%). And they argue that the formation of thrombus and synthetic steroid estrogen, estradiol, and further lead to coronary heart disease, further increases the risk of HSM [30]. In a phase III trial of ADT alone versus ADT combined with chemotherapy in the treatment of locally high-risk PC, no survival benefit was found with chemotherapy agents [31]. More seriously, Ahlgren et al. found in a prospective randomized study that docetaxel after chemotherapy in patients with PC had a numerically lower prognosis than the control group [32]. Although not statistically significant, the authors suggest that even in some patients chemotherapy drugs can cause PC to progress faster.

Cardiotoxicity after chemotherapy has been mentioned in a number of cancer-related studies, and the main symptoms include arrhythmia, acute coronary syndrome, diastolic dysfunction, cardiomyopathy and arterial hypertension [33]. A prospective study of 427 breast

cancer patients undergoing surgery found that patients receiving adjuvant chemotherapy with epirubicin/cyclophosphamide-docetaxel had a significantly higher incidence of CVD [34]. Similarly, Jin et al. conducted a retrospective analysis of patients with locally advanced or metastatic gastric cancer and found that the incidence of cardiac toxicity increased after receiving fluorouracil chemotherapy [35].

Chemotherapy combined with ADT has been the main systemic therapy for metastatic prostate cancer in the past decade. Most patients with locally advanced and metastatic prostate cancer receive gonadotropin-releasing hormone (GnRH) agonists in combination with docetaxel as recommended first-line treatment. Despite its survival benefits, it is associated with significant adverse effects, including sexual dysfunction, gynecomastia, anemia, osteoporosis, and cardiovascular disease [36]. Studies have shown that ADT combined with chemotherapy is positively correlated with the occurrence of cardiovascular disease (CVD) [37]. CVD accounts for 33% of annual deaths in the United States [38], compared with 45% in Europe [39]. In a recent study using the SEER database and the Swedish Cancer Registry, CVD was the most common cause of death in PC patients who survived more than 10 years after diagnosis [40]. Based on the "double-edged sword" effect of chemotherapy drugs in prostate cancer, it is very important to accurately determine whether chemotherapy can benefit PC patients with survival through big data.

Whether adjuvant chemotherapy can benefit the survival of elderly patients with PC has not been reported. Ibrahim et al. found that adriamycin based adjuvant chemotherapy was well tolerated in elderly breast cancer patients with good performance status and normal cardiac ejection fraction [41]. It is confirmed that patients with good cardiac function are generally less affected by the cardiotoxic side effects of chemotherapy drugs. In parallel, a multi-institutional population-based survival and toxicity analysis was conducted in patients with diffuse large B-cell lymphoma (DLBCL). The results show that chemotherapy can improve the survival rate of elderly DLBCL patients, and bring early survival benefits to elderly patients, without significantly increasing the risk of severe toxicity [42].

PC is a prevalent malignant tumor in men, and chemotherapy plays a significant role in its treatment, particularly in the management of castration-resistant prostate cancer (CRPC). Currently, the potential mechanisms by which chemotherapy contributes to long-term survival benefits for patients with prostate cancer (PC) are as follows: 1) Anti-mitotic effects: Chemotherapeutic agents such as docetaxel and cabazitaxel inhibit microtubule depolymerization, preventing cancer cell mitosis and thereby suppressing tumor cell proliferation. This anti-mitotic action is a direct mechanism by which chemotherapeutic drugs extend survival [43]. 2) Downregulation of androgen receptor (AR) activity: Docetaxel not only inhibits microtubule depolymerization but also downregulates the transcriptional activity of AR, suppresses AR gene expression, and inhibits AR activity by increasing FOXO1 levels [44]. This mechanism helps maintain therapeutic effects in CRPC [45]. 3) Promotion of apoptosis: Chemotherapeutic agents like docetaxel have anti-Bcl-2 and anti-Bcl-XL properties, exerting anti-tumor effects by promoting apoptosis [46]. The increase in apoptosis helps reduce tumor burden, potentially extending patient survival [47]. In addition to the three main mechanisms mentioned above, the anti-angiogenic and immunomodulatory effects of chemotherapeutic drugs should not be overlooked [48,49]. These effects make chemotherapy play a crucial role in PC, especially in CRPC, effectively prolonging patient survival.

In this study, we used data from the SEER database of elderly PC patients. We used advanced statistical methods, including propensity score matching and competitive risk models, to determine the effect of chemotherapy on patient heart-specific mortality. Our study showed that chemotherapy, age, T stage, and marriage were independent risk factors

for HSM. The results showed that chemotherapy did not increase the risk of HSM in patients. At the same time, we found that chemotherapy can improve the prognosis and prolong the survival time of elderly patients with PC compared with those who did not receive chemotherapy but had the same other conditions (P=0.002), which is consistent with relevant studies [21–25]. In addition, age, race, TNM stage, tumour tissue grade, PSA and Gleason score were independent influencing factors for patients receiving chemotherapy. The results show that the tumour itself and the basic condition of the patient are the keys to whether the patient receives chemotherapy.

Although strict statistical methods were used in this study, there are still some deficiencies in our study. Firstly, although propensity score matching was utilized to balance confounding factors such as age, race, tumor histological grade, TNM staging, surgery, radiotherapy, PSA levels, and Gleason scores, the matching process may not have entirely mitigated all potential confounding variables. Moreover, the reliance on retrospective data introduces the possibility of information bias, such as incomplete records or errors in data entry. Secondly, while sensitivity analyses were conducted to refine the matching outcomes, these analyses may not have captured all variables that could influence the results. Additionally, the SEER database's lack of patient comorbidity data means that, despite the use of propensity score matching to minimize the impact of confounding factors, the inherent effects of comorbidities on outcomes cannot be fully excluded. Lastly, the follow-up data for patients did not include the occurrence of complications, and data on chemotherapy-induced cardiotoxicity leading to heart disease were unavailable, potentially interfering with some of the study's findings. However, our study enhanced the PSM method to more accurately balance confounding factors between chemotherapy and non-chemotherapy groups. We also performed additional sensitivity analyses including different matching ratios (1:2 and 1:4) and IPTW using treatment weights to assess the robustness of our results under different assumptions. In addition, we performed stratified analyses based on key confounders to ensure that our findings were consistent across subgroups. Our CRR model now adjusts for a wider range of clinical and pathological factors, as well as treatment-related variables, to provide a more comprehensive assessment of the independent effects of chemotherapy on HSM. Therefore, our conclusion has a high degree of feasibility. We also focused on the potential impact of Lead Time Bias (LTB). LTB refers to the artifact of observed prolonged survival due to early diagnosis, which may affect the assessment of treatment effect. In order to ensure the accuracy and reliability of our findings, we used the uniform diagnosis time and follow-up time in the SEER database for our analysis, including the use of internal controls to avoid this bias [50]. At the same time, the use of PSM can balance confounding factors between screened and unscreened individuals, including the propensity to be screened, thereby reducing LTB[51].Therefore, further multicenter prospective clinical randomized controlled trials are necessary to validate our results, thereby enhancing the reliability and generalizability of the study's conclusions.

## Conclusion

We used propensity matching to analyze the incidence of HSM and overall death caused by chemotherapy in elderly patients with PC and reviewed the benefits and disadvantages of chemotherapy for patients with PC. For elderly patients with PC, HSM caused by CVD is an important cause of death. Doctors should not only pay attention to the risk of CSM of patients, but also monitor the risk of non-cancer death of patients during treatment. For elderly patients with PC, clinicians should rationally use chemotherapy drugs and adjust the treatment plan in time to achieve the purpose of prolonging the survival time of elderly patients with PC.

## Supporting information

**S1 File. Data.**
(CSV)

**S2 File. Raw data.**
(CSV)

**S3 File. Text data-File description.**
(PDF)

## Acknowledgments

Not applicable.

## Author contributions

**Conceptualization:** Chenghao Zhanghuang, Huake Wang, Li Li, Zipeng Hao.

**Data curation:** Chenghao Zhanghuang, Huake Wang, Li Li, Jinrong Li, Zipeng Hao, Jiacheng Zhang.

**Formal analysis:** Chenghao Zhanghuang, Huake Wang, Li Li, Jinrong Li.

**Funding acquisition:** Chenghao Zhanghuang, Li Li.

**Investigation:** Chenghao Zhanghuang, Jinkui Wang.

**Methodology:** Chenghao Zhanghuang, Jinkui Wang.

**Software:** Jinkui Wang.

**Validation:** Chenghao Zhanghuang.

**Writing – original draft:** Chenghao Zhanghuang, Jinkui Wang.

**Writing – review & editing:** Chenghao Zhanghuang, Ling Liu, Bing Yan.

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
