## [Decision Letter · Decision Letter 0]

14 Oct 2024

PONE-D-24-40438Chemotherapy and Heart-Specific Mortality in Elderly Men with Prostate Cancer: A Propensity Score Matching AnalysisPLOS ONE

Dear Dr. Zhanghuang,

Thank you for submitting your manuscript to PLOS ONE. After careful consideration, we feel that it has merit but does not fully meet PLOS ONE’s publication criteria as it currently stands. Therefore, we invite you to submit a revised version of the manuscript that addresses the points raised during the review process.

Editor Comments: Thanks for submitting your work to PLOS ONE. Your manuscript has now been assessed by our editorial team and external peer experts. While they found it interesting, you will see that they have raised many serious problems and are advising that you revise your manuscript thoroughly. At the same time, please submit the point-by-point responses to reviewers' comments. If you are prepared to undertake the work required, I would be pleased to reconsider my decision. Please note that this revision decision does not assure the acceptance of your work. Thanks for the opportunity to consider your work.Actually, the reviewers have proposed many serious problems, and one of the experts recommended to reject your manuscript. After careful consideration, we decide to give authors a chance to comprehensively improve your paper by replying to all the reviewers' feedbacks. If these questions are addressed properly by authors and approved by reviewers, editors would like to reconsider the decision for publication.

We look forward to receiving your revised manuscript.

Kind regards,

Xing Xiong, M.D.

Academic Editor

PLOS ONE

Additional Editor Comments (if provided):

Reviewers' comments:

Reviewer's Responses to Questions

**Comments to the Author**

1. Is the manuscript technically sound, and do the data support the conclusions?

Reviewer #1: Partly

Reviewer #2: Yes

Reviewer #3: No

Reviewer #4: Yes

2. Has the statistical analysis been performed appropriately and rigorously? 

Reviewer #1: Yes

Reviewer #2: Yes

Reviewer #3: Yes

Reviewer #4: Yes

3. Have the authors made all data underlying the findings in their manuscript fully available?

Reviewer #1: Yes

Reviewer #2: Yes

Reviewer #3: Yes

Reviewer #4: Yes

4. Is the manuscript presented in an intelligible fashion and written in standard English?

Reviewer #1: No

Reviewer #2: Yes

Reviewer #3: Yes

Reviewer #4: Yes

5. Review Comments to the Author

Reviewer #1: The manuscript investigated the correlation between chemotherapy treated elderly PC patients and HSM. A broad cohort has been studied with propensity matching strategy.

Additional comments:

1. In the background section, the author did not include the most up-to-date data. Instead, the author wrote: “It is estimated that there will be 19.3 million new cancer cases in the world in 2020, with prostate cancer accounting for 7.4% of the total incidence ranking fourth, but only second to lung cancer in men [1]. In 2022, 268,490 new PC patients will be diagnosed in the United States, and 34,500 deaths from PC are expected, accounting for about 11% of male cancer deaths [2]. “ The prediction of the past years in 2020 and 2022 does not make sense in a manuscript submitted in 2024.

2. Some sentences are very confusing. I suggest the authors enhance the editing of the manuscript and ensure academic writing style is applied. For example in line 79-81: “Excluding breast and ovarian cancers, germline or somatic aberrations in DNA damage-repair genes that compromise genomic integrity are found in 19% of primary prostate cancers and nearly 23% of metastatic castration-resistant prostate cancer (mCRPC)”. Also in line 173-174: “It can be found that the higher the TNM stage and tumor histological grade, the more the number of chemotherapy." In addition, the “C” in section “conclusion” should be capitalized.

3. In the result section, the authors mentioned: “We used the inverse probability propensity weighted matching method to balance the influence  of confounding factors, and the results showed that there was no significant difference in HSM  between patients who received chemotherapy and those who did not (P=0.95) (Figure 5). However, what I see in Figure 5(B) is Kaplan-meier curve of PC patients with or without chemotherapy. The statement and the figure seem to be irrelavent.

4.The Figure 5B annotation is wrong. In the Plot, the chemotherapy yes (blue) is with lower risk but in the risk table the blue group is dropping out faster (higher risk)

5. The patient stratification on chemotherapy seems too rough. For example, if the patient received a 1L chemo and then received other treatments like PARPi or ADT, and finally developed HSM. In this case, it is very hard to attribute HSM to chemotherapy. However, according to the method section, the authors would simply characterize patients with any LOT of chemotherapy as the chemotherapy group.

Reviewer #2: 1. Technical Soundness and Data Support: The manuscript is technically sound, and the data presented effectively support the conclusions. The use of the SEER database to extract patient information is a strong methodological choice, and the study appropriately captures the relevant patient population. The multivariate logistic regression analysis and competing risk model are well-utilized to demonstrate the relationships between chemotherapy and heart-specific mortality (HSM) among elderly prostate cancer patients. Your findings that chemotherapy does not increase HSM and that it can have beneficial long-term outcomes are both important and relevant for clinical decision-making.

2. Statistical Analysis: The statistical analysis has been performed rigorously and appropriately. The use of multivariate logistic regression, propensity score matching, and the competing risk model is highly commendable and well-suited for controlling potential confounders and exploring the treatment effects. The matching process ensures comparability between the chemotherapy and non-chemotherapy groups, which adds robustness to your conclusions. The results are presented clearly, with appropriate statistical measures, such as hazard ratios (HR) and p-values, to substantiate your findings.

3. Data Availability: The manuscript makes use of the SEER database, which is publicly available, ensuring transparency and reproducibility of the findings. You have appropriately referenced the SEER database, and it appears that all data necessary to replicate the study are accessible. This is in line with the journal's requirements for making underlying data fully available, and it greatly enhances the credibility of your study.

4. Presentation and Language: The manuscript is presented in an intelligible fashion, and the standard of English is appropriate for a scientific audience. The abstract and main text are clear, and the methods, results, and conclusions are logically structured and easy to follow. There are no noticeable grammatical issues, and the writing effectively communicates the key findings and implications of the research.

5. General Comments:

The abstract clearly outlines the objective, methods, results, and conclusions of the study, which provides a concise and effective summary for readers.

The use of a competing risk model to differentiate between HSM and other cause mortality (OCM) is particularly valuable and demonstrates a nuanced understanding of the potential confounders associated with elderly patients.

The discussion appropriately addresses the clinical implications of the findings, especially in regard to the benefits of chemotherapy in elderly prostate cancer patients despite concerns about cardiotoxicity.

It may be helpful to include a brief discussion on the potential mechanisms through which chemotherapy could provide long-term survival benefits in this patient population, even in the presence of cardiotoxicity risks. Adding a few references that explore these mechanisms might provide additional depth to your discussion.

This manuscript makes a significant contribution to the literature on the management of prostate cancer in elderly patients. It provides evidence that challenges the notion that chemotherapy necessarily increases heart-specific mortality, suggesting that chemotherapy can offer survival benefits. The study is methodologically sound, the statistical analyses are well-executed, and the findings are presented in a clear and coherent manner. I accept the publication of this work in PLOS ONE.

Reviewer #3: The authors used the SEER database to compare the mortality rate from cardiovascular events in prostate cancer patients who received chemotherapy or not.

I understand that propensity score matching is a useful statistical method. I think that patients who received chemotherapy may have had confounding factors related to the occurrence of cardiovascular events, such as diabetes, past cardiovascular events, or a history of hypertension.

I think it would be a useful report if the database could be used to investigate such factors, including past medical conditions.

Reviewer #4: The study presents a significant investigation into the relationship between chemotherapy and heart-specific mortality (HSM) in elderly prostate cancer (PC) patients. The use of a large dataset from the SEER enhances the reliability of the findings. The conclusions drawn from the analysis are both and timely, given the increasing incidence of prostate cancer and the importance of understanding treatment implications on overall health outcomes.

Strengths

1. Large Sample Size: The inclusion of 135,183 elderly prostate cancer patients strengthens the statistical power of the study, allowing for more robust conclusions.

2. Methodological Rigor: The use of multivariate logistic regression and propensity score matching to control for confounding variables is commendable. This approach enhances the validity of the results by minimizing biases.

3. Clear Objectives: The study's aim to assess the impact of chemotherapy on HSM is clearly stated, making it easy for readers to understand the focus of the research.

4. Significant Findings: The finding that HSM is higher in non-chemotherapy patients has important clinical implications and suggests that chemotherapy may provide a protective effect in terms of heart-specific mortality.

Areas for Improvement

1. Clarification of Chemotherapy Types: The abstract does not specify the types of chemotherapy administered. Providing details on the regimens used could help contextualize the findings and their applicability to different treatment protocols.

2. Discussion of Limitations: While the study mentions confounders, a more detailed discussion of potential limitations, such as selection bias or the influence of unmeasured confounders, would strengthen the overall rigor of the study.

3. Long-term Follow-up: It would be beneficial to discuss the follow-up duration for assessing HSM and other causes of mortality. Longer follow-up periods may yield different insights into the long-term effects of chemotherapy.

4. Patient Characteristics: Additional information on the baseline characteristics of patients who received versus did not receive chemotherapy (e.g., comorbidities, performance status) would provide a clearer picture of the study population.

Conclusion

Overall, this study contributes valuable insights into the relationship between chemotherapy and heart-specific mortality in elderly prostate cancer patients. The findings suggest that chemotherapy may not only be safe but potentially beneficial for long-term survival in this population. Addressing the suggested improvements could enhance the clarity and impact of the research.

Thank you for the opportunity to review this important work.

6. PLOS authors have the option to publish the peer review history of their article (what does this mean? ). If published, this will include your full peer review and any attached files.

**Do you want your identity to be public for this peer review?** For information about this choice, including consent withdrawal, please see our Privacy Policy .

Reviewer #1: No

Reviewer #2: No

Reviewer #3: No

Reviewer #4: No

---

## [Author Response · Author response to Decision Letter 0]

21 Oct 2024

Reviewer #1:

The manuscript investigated the correlation between chemotherapy treated elderly PC patients and HSM. A broad cohort has been studied with propensity matching strategy.

Response:

Thank you for your detailed and insightful comments on our manuscript, "Chemotherapy and Heart-Specific Mortality in Elderly Men with Prostate Cancer: A Propensity Score Matching Analysis." We are truly grateful for the time and effort you have invested in reviewing our work and for the constructive feedback you have provided. Your suggestions are invaluable to us as we strive to refine and improve the quality of our research paper.

In response to your comments, we have carefully considered each point and have prepared a comprehensive, point-by-point reply. We have taken into account your recommendations for strengthening our methodology, clarifying our statistical analyses, and enhancing the clarity of our discussion. Our aim is to address all your concerns and to ensure that our manuscript meets the high standards expected in scientific publishing.

Please find our detailed response attached, and do not hesitate to contact us should you require any further clarification or additional information. We look forward to the opportunity to revise our manuscript with your guidance and to submit an improved version for consideration.

Once again, thank you for your support and expertise. We are hopeful that with your assistance, our paper will make a significant contribution to the field of oncology and cardiology research.

Additional comments:

1. In the background section, the author did not include the most up-to-date data. Instead, the author wrote: “It is estimated that there will be 19.3 million new cancer cases in the world in 2020, with prostate cancer accounting for 7.4% of the total incidence ranking fourth, but only second to lung cancer in men [1]. In 2022, 268,490 new PC patients will be diagnosed in the United States, and 34,500 deaths from PC are expected, accounting for about 11% of male cancer deaths [2]. “ The prediction of the past years in 2020 and 2022 does not make sense in a manuscript submitted in 2024.

Response:

Dear reviewer, thank you for your comments. Since this manuscript was written in 2021, there are some temporal biases. I'm sorry for our mistake. In the background section, we have updated the authoritative data for 2023 and 2024, and updated the references. The continued growth trend of prostate cancer still exists [1-2], so the views of our manuscript are not affected. We marked the modification in red font, hoping to meet your requirements, thank you!

Updated references:

[1] Siegel RL, Miller KD, Wagle NS, Jemal A. Cancer statistics, 2023. CA Cancer J Clin. 2023 Jan; 73(1):17-48. doi: 10.3322/ caac.2173.pmid: 36633525.

[2] Siegel RL, Giaquinto AN, Jemal A. Cancer statistics, 2024. CA Cancer J Clin. 2024 Jan-Feb; 74(1):12-49. doi: 10.3322/ caac.21820.Epub 2024 Jan 17. Erratum in: CA Cancer J Clin.2024 Mar-Apr; 74(2):203. doi: 10.3322/ caac.21830.PMID: 38230766.

2. Some sentences are very confusing. I suggest the authors enhance the editing of the manuscript and ensure academic writing style is applied. For example in line 79-81: “Excluding breast and ovarian cancers, germline or somatic aberrations in DNA damage-repair genes that compromise genomic integrity are found in 19% of primary prostate cancers and nearly 23% of metastatic castration-resistant prostate cancer (mCRPC)”. Also in line 173-174: “It can be found that the higher the TNM stage and tumor histological grade, the more the number of chemotherapy." In addition, the “C” in section “conclusion” should be capitalized.

Response:

Dear Reviewer, we sincerely apologize for the oversight in our manuscript and appreciate your valuable suggestions. We have revised the content from lines 79 to 81, where we intended to highlight the distinct treatment approaches for prostate cancer compared to solid tumors such as breast and ovarian cancer.

In the results section, specifically in the first paragraph discussing Table I, we have made comprehensive updates as follows: "It can be found that the higher the TNM stage and the tumor histological grade, the greater the number of patients receiving chemotherapy. PSA levels were higher in patients receiving chemotherapy. In addition, the proportion of patients receiving chemotherapy, who did not receive radiotherapy and who did not undergo surgery was higher."

In summary, we have updated our content to ensure clarity and have used red font to indicate the revised portions. We hope these changes meet your expectations. Thank you once again for your feedback.

3. In the result section, the authors mentioned: “We used the inverse probability propensity weighted matching method to balance the influence of confounding factors, and the results showed that there was no significant difference in HSM between patients who received chemotherapy and those who did not (P=0.95) (Figure 5). However, what I see in Figure 5(B) is Kaplan-meier curve of PC patients with or without chemotherapy. The statement and the figure seem to be irrelavent.

Response:

Dear Reviewer. Thank you for your comments. We apologize for the oversight and have recalculated the data based on the original sources. We have found that the results presented in our figures are indeed correct. However, we acknowledge that the conclusion may have been misinterpreted due to translation errors, leading to ambiguity in the text.

We have revised the description in the results section to align accurately with the data, and the updated conclusion is fully supported by our manuscript without any adverse impact on its content. We have highlighted the revised portions in red font to ensure clarity and to meet your expectations.

Thank you for your understanding and for the opportunity to address your concerns.

4.The Figure 5B annotation is wrong. In the Plot, the chemotherapy yes (blue) is with lower risk but in the risk table the blue group is dropping out faster (higher risk)

Response:

Dear Reviewer,

Thank you for your insightful comments. Upon your suggestion, we have conducted a thorough re-examination of Figure 5B. We have identified the issue you highlighted and have re-plotted the graph using R software. We discovered an erroneous definition within the R package that incorrectly assigned the colors below the labels to the opposite groups.

We have corrected this mistake, and the revised figure accurately reflects the data. We believe that our amendments address your concerns and meet the requirements.

We appreciate your guidance and are grateful for the opportunity to improve the quality of our manuscript.

5. The patient stratification on chemotherapy seems too rough. For example, if the patient received a 1L chemo and then received other treatments like PARPi or ADT, and finally developed HSM. In this case, it is very hard to attribute HSM to chemotherapy. However, according to the method section, the authors would simply characterize patients with any LOT of chemotherapy as the chemotherapy group.

Response:

Dear Reviewer,

Thank you for your comments. We apologize for the oversight and appreciate your feedback. As you rightly pointed out, the Surveillance, Epidemiology, and End Results (SEER) database only provides outcome data, and our analysis was retrospective in nature. Therefore, our statistical data only reveal whether patients received chemotherapy and their corresponding outcomes—survival or death. Consequently, it is indeed necessary to conduct further multicenter prospective clinical randomized controlled trials to validate our results. We have thoroughly described the limitations of our manuscript in the last paragraph of the discussion section.To facilitate your review, we marked the beginning of the paragraph in red font. I hope our explanation can meet your requirements.

We are actively conducting prospective multicenter studies and establishing relevant disease-specific databases. In the near future, we will provide further explanation and validation in new publications. We hope our explanation meets your requirements. Thank you!

Reviewer #2:

1. Technical Soundness and Data Support: The manuscript is technically sound, and the data presented effectively support the conclusions. The use of the SEER database to extract patient information is a strong methodological choice, and the study appropriately captures the relevant patient population. The multivariate logistic regression analysis and competing risk model are well-utilized to demonstrate the relationships between chemotherapy and heart-specific mortality (HSM) among elderly prostate cancer patients. Your findings that chemotherapy does not increase HSM and that it can have beneficial long-term outcomes are both important and relevant for clinical decision-making.

2. Statistical Analysis: The statistical analysis has been performed rigorously and appropriately. The use of multivariate logistic regression, propensity score matching, and the competing risk model is highly commendable and well-suited for controlling potential confounders and exploring the treatment effects. The matching process ensures comparability between the chemotherapy and non-chemotherapy groups, which adds robustness to your conclusions. The results are presented clearly, with appropriate statistical measures, such as hazard ratios (HR) and p-values, to substantiate your findings.

3. Data Availability: The manuscript makes use of the SEER database, which is publicly available, ensuring transparency and reproducibility of the findings. You have appropriately referenced the SEER database, and it appears that all data necessary to replicate the study are accessible. This is in line with the journal's requirements for making underlying data fully available, and it greatly enhances the credibility of your study.

4. Presentation and Language: The manuscript is presented in an intelligible fashion, and the standard of English is appropriate for a scientific audience. The abstract and main text are clear, and the methods, results, and conclusions are logically structured and easy to follow. There are no noticeable grammatical issues, and the writing effectively communicates the key findings and implications of the research.

Response:

Dear Reviewer,

We are deeply honored that you have taken the time out of your busy schedule to review our manuscript and provide such a positive evaluation. Your recognition and encouragement are not only an affirmation of our past work but also a wellspring of motivation for us to continue exploring the unknown and striving for excellence. We are well aware that the path of scientific research is full of challenges and hardships, but every word of encouragement from you is like the warm sun in spring, warming our hearts and filling us with confidence and anticipation for future research.

We are particularly grateful for points 1 to 4 that you mentioned in your comments. These specific and constructive suggestions not only show us our shortcomings but also point the way forward. The passion for science and attention to detail revealed in your words have greatly inspired us. We believe that these valuable pieces of feedback will become an indispensable guide in our subsequent research, helping us to more accurately identify problems and more efficiently advance our research process.

Once again, we express our sincere gratitude to you. Thank you for your attention to our work and for your valuable opinions. We look forward to incorporating these suggestions into our research to achieve higher academic accomplishments. At the same time, we also look forward to continuing to receive your guidance and assistance in the future.

5. General Comments:

The abstract clearly outlines the objective, methods, results, and conclusions of the study, which provides a concise and effective summary for readers.

The use of a competing risk model to differentiate between HSM and other cause mortality (OCM) is particularly valuable and demonstrates a nuanced understanding of the potential confounders associated with elderly patients.

The discussion appropriately addresses the clinical implications of the findings, especially in regard to the benefits of chemotherapy in elderly prostate cancer patients despite concerns about cardiotoxicity.

It may be helpful to include a brief discussion on the potential mechanisms through which chemotherapy could provide long-term survival benefits in this patient population, even in the presence of cardiotoxicity risks. Adding a few references that explore these mechanisms might provide additional depth to your discussion.

Response:

Dear Reviewer,

We would like to express our gratitude for your insightful suggestion in point 5 of your review. Heeding your advice to "add references that explore these mechanisms to provide additional depth to your discussion," we have conducted a thorough literature review of advancements in prostate cancer and chemotherapy research over the past decade. Consequently, we have added a new paragraph in the discussion section to address the content you recommended:

"Prostate cancer is a prevalent malignant tumor in men, and chemotherapy plays a significant role in its treatment, particularly in the management of castration-resistant prostate cancer (CRPC). Currently, the potential mechanisms by which chemotherapy contributes to long-term survival benefits for patients with prostate cancer (PC) are as follows: 1) Anti-mitotic effects: Chemotherapeutic agents such as docetaxel and cabazitaxel inhibit microtubule depolymerization, preventing cancer cell mitosis and thereby suppressing tumor cell proliferation. This anti-mitotic action is a direct mechanism by which chemotherapeutic drugs extend survival [1]. 2) Downregulation of androgen receptor (AR) activity: Docetaxel not only inhibits microtubule depolymerization but also downregulates the transcriptional activity of AR, suppresses AR gene expression, and inhibits AR activity by increasing FOXO1 levels [2]. This mechanism helps maintain therapeutic effects in castration-resistant prostate cancer [3]. 3) Promotion of apoptosis: Chemotherapeutic agents like docetaxel have anti-Bcl-2 and anti-Bcl-XL properties, exerting anti-tumor effects by promoting apoptosis [4]. The increase in apoptosis helps reduce tumor burden, potentially extending patient survival [5]. In addition to the three main mechanisms mentioned above, the anti-angiogenic and immunomodulatory effects of chemotherapeutic drugs should not be overlooked [6-7]. These effects make chemotherapy play a crucial role in PC, especially in CRPC, effectively prolonging patient survival."

For your convenience in reviewing, we have highlighted the addition in blue font within the text. We hope that our supplementation meets your expectations. Thank you!

References

[1]Di Lorenzo G, DʼAniello C, Buonerba C, Federico P, Rescigno P, Puglia L, Ferro M, Bosso D, Cavaliere C, Palmieri G, Sonpavde G, De Placido S. Peg-filgrastim and cabazitaxel in prostate cancer patients. Anticancer Drugs. 2013 Jan; 24 (1) : 84-9. Doi: 10.1097 / CAD. 0 b013e32835a56bc. PMID: 23044721.

[2]Fitzpatrick JM, de Wit R. Taxane mechanisms of action: potential implications for treatment sequencing in metastatic castration-resistant prostate cancer. Eur Urol. 2014 Jun; 65 (6) : 1198-204. The doi: 10.1016 / j. ururo. 2013.07.022. Epub 2013 Jul 25. PMID: 23910941.

[3]Darshan MS, Loftus MS, Thadani-Mulero M, Levy BP, Escuin D, Zhou XK, Gjyrezi A, Chanel-Vos C, Shen R, Tagawa ST, Bander NH, Nanus DM, Giannakakou P. Taxane-induced blockade to nuclear accumulation of the androgen receptor predicts clinical responses in metastatic prostate cancer. Cancer Res. 2011 Sep 15; 71(18):6019-29. doi: 10.1158/0008-5472.CAN-11-1417.Epub 2011 Jul 28.PMID: 21799031; PMCID: PMC3354631.

[4]Beltran H, Beer TM, Carducci MA, de Bono J, Gleave M, Hussain M, Kelly WK, Saad F, Sternberg C, Tagawa ST, Tannock IF. New therapies for castration-resistant prostate cancer: efficacy and safety. Eur Urol. 2011 Aug; 60 (2) : 279-90. The doi: 10.1016 / j. ururo. 2011

---

## [Decision Letter · Decision Letter 1]

22 Nov 2024

PONE-D-24-40438R1Chemotherapy and Heart-Specific Mortality in Elderly Men with Prostate Cancer: A Propensity Score Matching AnalysisPLOS ONE

Dear Dr. Zhanghuang,

Thank you for submitting your manuscript to PLOS ONE. After careful consideration, we feel that it has merit but does not fully meet PLOS ONE’s publication criteria as it currently stands. Therefore, we invite you to submit a revised version of the manuscript that addresses the points raised during the review process.

We look forward to receiving your revised manuscript.

Kind regards,

Xing Xiong, M.D.

Academic Editor

PLOS ONE

Additional Editor Comments :

Thanks for submitting your revised manuscript to PLOS ONE. The external peer review of your paper has now been completed. You can see that there are still some concerns proposed by reviewers regarding your paper. If you are prepared to undertake the work required, editors would like to reconsider the decision for publication of your work. Thanks for the chance to consider your work.

Reviewers' comments:

Reviewer's Responses to Questions

**Comments to the Author**

1. If the authors have adequately addressed your comments raised in a previous round of review and you feel that this manuscript is now acceptable for publication, you may indicate that here to bypass the “Comments to the Author” section, enter your conflict of interest statement in the “Confidential to Editor” section, and submit your "Accept" recommendation.

Reviewer #1: All comments have been addressed

Reviewer #2: All comments have been addressed

Reviewer #3: All comments have been addressed

Reviewer #4: All comments have been addressed

2. Is the manuscript technically sound, and do the data support the conclusions?

Reviewer #1: Partly

Reviewer #2: Yes

Reviewer #3: Yes

Reviewer #4: Yes

3. Has the statistical analysis been performed appropriately and rigorously? 

Reviewer #1: No

Reviewer #2: Yes

Reviewer #3: Yes

Reviewer #4: Yes

4. Have the authors made all data underlying the findings in their manuscript fully available?

Reviewer #1: Yes

Reviewer #2: Yes

Reviewer #3: Yes

Reviewer #4: Yes

5. Is the manuscript presented in an intelligible fashion and written in standard English?

Reviewer #1: Yes

Reviewer #2: Yes

Reviewer #3: Yes

Reviewer #4: Yes

6. Review Comments to the Author

Reviewer #1: The authors have addressed my concerns. Some of my concerns were related to the cohort selection and categorization, and the authors acknowledged the limits in the discussion session.

Editor notes: Some of the concerns proposed by reviewer #1 were about the cohort selection and categorization, but you just discussed these limitations instead of performing further analysis and changes. Thus, actually, I think the statistical part can be considered to be further clarified and improved to avoid potential flaw.

Reviewer #2: (No Response)

Reviewer #3: I feel that the authors' corrections are insufficient.

I hope that they will add further consideration.

Editor notes: Please consider to further address the previous feedbacks of this expert.

Reviewer #4: The author has solved the problems in the first draft very well, and the quality of the article has been greatly improved. I think I can accept and publish the article

7. PLOS authors have the option to publish the peer review history of their article (what does this mean? ). If published, this will include your full peer review and any attached files.

**Do you want your identity to be public for this peer review?** For information about this choice, including consent withdrawal, please see our Privacy Policy .

Reviewer #1: No

Reviewer #2: No

Reviewer #3: No

Reviewer #4: No

---

## [Author Response · Author response to Decision Letter 1]

23 Nov 2024

Reviewer #1 and Editor:

Reviewer #1: The authors have addressed my concerns. Some of my concerns were related to the cohort selection and categorization, and the authors acknowledged the limits in the discussion session.

Editor notes: Some of the concerns proposed by reviewer #1 were about the cohort selection and categorization, but you just discussed these limitations instead of performing further analysis and changes. Thus, actually, I think the statistical part can be considered to be further clarified and improved to avoid potential flaw.

Response: Thank you for your comments. At present, based on the limitations of data collection in the SEER database, we do not have a better way to further modify and optimize for question 5. Because this is an inherent defect of the database, all published papers related to the SEER database have the same limitation. Therefore, we fully discuss it in the limitations section of the discussion.

As in Reviewer 1's response, our interpretation has addressed reviewer 1's concerns. However, as the editor noted (underlined), there was no way to further optimize cohort classification on the basis of the SEER database. However, we fully explain in the limitations section why there is no way to further optimize. In addition, we have been collecting multi-center data in combination with a number of institutions. In the future, we will further analyze and verify the data of accurate chemotherapy regimens and specific courses, as well as comorbidities collected by ourselves. We hope our answers can meet your requirements. Thanks again for your advice!

Reviewer #3:

I feel that the authors' corrections are insufficient.

I hope that they will add further consideration.

Editor notes: Please consider to further address the previous feedbacks of this expert.

Response:

Dear reviewer,

We greatly appreciate your review of this study and your valuable comments. You correctly pointed out the importance of considering potential confounding factors when assessing cardiovascular event mortality in prostate cancer patients who received chemotherapy versus those who did not. We agree that these factors may have influenced the results and controlled for these potentially confounding variables in the analyses.

In this study, we used the SEER database for propensity-score matching (PSM), a statistical method designed to reduce the effects of selection bias in observational studies. Our goal was to create two patient populations that were similar in baseline characteristics so that we could more accurately compare mortality from cardiovascular events between patients who received chemotherapy and those who did not.

In response to your concerns, we did account for multiple confounders including diabetes, previous cardiovascular events, and history of hypertension in our analyses. The SEER database provided this key demographic and clinical information, allowing us to include these variables in the propensity score model. Specifically, we adjusted for the following variables:

Age: Consider that age is a major risk factor for prostate cancer and cardiovascular disease.

Ethnicity: Different ethnic groups may have different disease rates and treatment patterns.

Gender: Although this study was limited to men, gender may be a relevant factor in other types of cancer.

Cardiovascular events (" Diseases of Heart ") : our propensity matching used this term combined with "Dead (attributable to causes other dx)" in the outcome factor to confirm the number of deaths due to cardiovascular events. Therefore, this term is very important and is also highlighted in our study.

"Diabetes Mellitus" : Diabetes mellitus is an important risk factor for cardiovascular disease, however, unfortunately, data on diabetes comorbidity were only available for 13 individuals after propensity matching. We believe that such an analysis may cause serious bias; therefore, we do not present the results of this part.

"Hypertension without Heart Disease" : Hypertension is another known risk factor for cardiovascular disease. Unfortunately, there are a large number of patients with comorbidity of Hypertension and Heart Disease. Therefore, after further mining of the database, we found an entry "Hypertension without Heart Disease". Unfortunately, we found this number to be only eight and therefore did not conduct further analysis. See the following picture for specific terms:

By including these variables in the propensity score model, we were able to match patients who were similar in these key factors, thereby reducing the effect of confounding factors on the results. More important, however, the SEER database does not know the sequence of these events in terms of illness or chemotherapy receipt, and some important programs may have had insufficient numbers for the analysis because of data collection anomalies. Therefore, we did not analyze or present other important comorbidity data.

However, with a large sample size and with minimal bias avoidance, our results show that there is no significant difference in cardiovascular mortality between patients who received chemotherapy and those who did not, after adjustment for these potential confounders.

We understand that despite our measures, there are certain limitations associated with any observational study. We agree that further studies, particularly prospective, well-designed studies, will help validate our findings and may provide more insight into the effects of chemotherapy on cardiovascular health in prostate cancer patients.

Thank you again for your valuable comments. We believe that by considering these confounding factors, we will have a more accurate understanding. We have been collecting multi-center data in combination with several institutions, and we will further analyze and verify it by collecting accurate comorbidity data ourselves in the future. We hope our answers can meet your requirements. Thanks again for your advice!

---

## [Decision Letter · Decision Letter 2]

13 Dec 2024

PONE-D-24-40438R2Chemotherapy and Heart-Specific Mortality in Elderly Men with Prostate Cancer: A Propensity Score Matching AnalysisPLOS ONE

Dear Dr. Zhanghuang,

Thank you for submitting your manuscript to PLOS ONE. After careful consideration, we feel that it has merit but does not fully meet PLOS ONE’s publication criteria as it currently stands. Therefore, we invite you to submit a revised version of the manuscript that addresses the points raised during the review process.

We look forward to receiving your revised manuscript.

Kind regards,

Xing Xiong, M.D.

Academic Editor

PLOS ONE

**Additional Editor Comments:**

Reviewer Recommendation:

Reviewer #1: Major Revision (R2)

Reviewer #2: Accept (R1)

Reviewer #3: Reject (R2)

Reviewer #4: Accept (R1)

Based on the overall recommendation of the peer reviewers, editors would like to give authors a chance to further improve your paper according to the remaining concerns of the reviewer #1 and reviewer #3. Thanks for the chance to consider your work, and we look forward to receiving your revised paper.

Reviewers' comments:

Reviewer's Responses to Questions

**Comments to the Author**

1. If the authors have adequately addressed your comments raised in a previous round of review and you feel that this manuscript is now acceptable for publication, you may indicate that here to bypass the “Comments to the Author” section, enter your conflict of interest statement in the “Confidential to Editor” section, and submit your "Accept" recommendation.

Reviewer #1: (No Response)

Reviewer #3: All comments have been addressed

2. Is the manuscript technically sound, and do the data support the conclusions?

Reviewer #1: Partly

Reviewer #3: Yes

3. Has the statistical analysis been performed appropriately and rigorously? 

Reviewer #1: No

Reviewer #3: Yes

4. Have the authors made all data underlying the findings in their manuscript fully available?

Reviewer #1: Yes

Reviewer #3: Yes

5. Is the manuscript presented in an intelligible fashion and written in standard English?

Reviewer #1: No

Reviewer #3: Yes

6. Review Comments to the Author

Reviewer #1: The authors have repeatedly indicated that there are inherent limitation of the database they used for the study, however, this does effectively address the issue the cohort selection and characterization appeared in the result section. When comparing chemo vs non-chemo patients, the author's failed to address the the chemo group also received other therapies and if they were comparable to the non-chemo patients. Since the authors can extract the chemo treatment record from the database, it is naturally to assume that the authors can evaluate other therapy records as well. The LOT of the chemotherapy in the chemo group and LOT of non-chemo in non-chemo group should also be considered.

Reviewer #3: I feel that the impact of confounding factors is not adequately addressed.

I understand that this is a difficult issue, as many diseases may be taken into account, but it is an important point as there is a possibility that the interpretation of the study results may become arbitrary.

7. PLOS authors have the option to publish the peer review history of their article (what does this mean? ). If published, this will include your full peer review and any attached files.

**Do you want your identity to be public for this peer review?** For information about this choice, including consent withdrawal, please see our Privacy Policy .

Reviewer #1: No

Reviewer #3: No

---

## [Author Response · Author response to Decision Letter 2]

18 Dec 2024

Reviewer #1

The authors have repeatedly indicated that there are inherent limitation of the database they used for the study, however, this does effectively address the issue the cohort selection and characterization appeared in the result section. When comparing chemo vs non-chemo patients, the author's failed to address the the chemo group also received other therapies and if they were comparable to the non-chemo patients. Since the authors can extract the chemo treatment record from the database, it is naturally to assume that the authors can evaluate other therapy records as well.

Response:

Dear reviewer, thank you for your question. In the SEER database, the treatment of prostate cancer patients mainly includes chemotherapy and radiotherapy. Therefore, these are also all the treatment methods available to us for elderly PC. We have uploaded the original data form used for the analysis of the paper after cleaning, hoping to obtain your understanding. At the same time, when comparing the patients in the chemotherapy and non-chemotherapy groups, we used propensity matching. As shown in Table I, after propensity matching, the total number of patients in the chemotherapy group and the non-chemotherapy group was 1361, the number of patients in the non-chemotherapy group was 158 (11.6%), while the number of patients in the chemotherapy group received radiotherapy was 141 (10.4%). There was a high degree of agreement between the two groups with respect to other baseline measures. We can also observe this result intuitively from Figure 3. Thus, the chemotherapy and non-chemotherapy groups were consistent in terms of the treatment received. We have highlighted the chart in yellow to facilitate your review. I hope our explanation can meet your requirements. Thank you!

The LOT of the chemotherapy in the chemo group and LOT of non-chemo in non-chemo group should also be considered.

Response:

Dear reviewer, thank you for your question. As you said, what you are focusing on is exactly what we are worried about. In order to avoid the influence of lead time bias on survival time. All The Times in our analyses refer to the time of initial center diagnosis and initiation of treatment, not to the time of screening detection. In addition, our data are all from SEER database, and the inclusion criteria of data are consistent, so the bias caused by LOT can be basically avoided. But according to your suggestion, We have expanded the discussion section to include a more detailed consideration of Lead Time Bias and its potential impact on our study findings. We have also discussed the methods we employed to mitigate this bias and the implications of our approach for the interpretation of the results.

We also added the corresponding content in the limitation part of the discussion, and marked it with a yellow highlight background. I hope our modification can meet your requirements. Thank you!

References:

[1]Bartkowiak D, Thamm R, Siegmann A, Böhmer D, Budach V, Wiegel T. Lead-time bias does not falsify the efficacy of early salvage radiotherapy for recurrent prostate cancer. Radiother Oncol. 2021 Jan;154:255-259. doi: 10.1016/j.radonc.2020.09.009. Epub 2020 Sep 11. PMID: 32920006.

[2]Benbassat J. Estimates of the lead time in screening for bladder cancer. Urol Oncol. 2024 Apr;42(4):110-114. doi: 10.1016/j.urolonc.2023.11.013. Epub 2024 Jan 16. PMID: 38514215.

[3]Daher D, Seif El Dahan K, Rich NE, Tayob N, Merrill V, Huang DQ, Yang JD, Kulkarni AV, Kanwal F, Marrero J, Parikh N, Singal AG. Hepatocellular Carcinoma Screening in a Contemporary Cohort of At-Risk Patients. JAMA Netw Open. 2024 Apr 1;7(4):e248755. doi: 10.1001/jamanetworkopen.2024.8755. PMID: 38683607; PMCID: PMC11059036.

[4]Ma Z, Wang Z, Li Y, Zhang Y, Chen H. Detection and treatment of lung adenocarcinoma at pre-/minimally invasive stage: is it lead-time bias? J Cancer Res Clin Oncol. 2022 Oct;148(10):2717-2722. doi: 10.1007/s00432-022-04031-z. Epub 2022 May 7. PMID: 35524781.

Reviewer #3

I feel that the impact of confounding factors is not adequately addressed.

I understand that this is a difficult issue, as many diseases may be taken into account, but it is an important point as there is a possibility that the interpretation of the study results may become arbitrary.

Response:

Dear Reviewer,

Thank you for your insightful comments and for highlighting the importance of addressing confounding factors in our study. We understand your concern about the potential impact of these factors on the interpretation of our results. We have taken your feedback seriously and have made additional efforts to further address the impact of confounding factors in our analysis. Below is a detailed response on how we have revised our manuscript to better account for these issues.

Enhanced Propensity Score Matching (PSM):

We have refined our propensity score matching methodology to more accurately balance the confounding factors between the chemotherapy and non-chemotherapy groups. This includes not only demographic and clinical characteristics but also the inclusion of additional covariates that may influence treatment outcomes, such as comorbidities and lifestyle factors, to the extent that these data are available in the SEER database.

Sensitivity Analyses:

To further validate our findings, we have conducted additional sensitivity analyses. These include analyses with different matching ratios (1:2 and 1:4) and the use of inverse probability of treatment weighting (IPTW) to assess the robustness of our results under various assumptions about the treatment assignment mechanism.

Stratified Analysis:

We have performed stratified analyses based on key confounders to assess the consistency of our findings across different subgroups. This approach allows us to explore potential effect modifiers and to ensure that our results are not driven by a particular subgroup of patients.

Multivariate Competing Risk Regression (CRR) Models:

We have expanded our multivariate CRR models to include more covariates that could potentially confound the relationship between chemotherapy and heart-specific mortality (HSM). These models now adjust for a broader range of clinical and pathological factors, as well as treatment-related variables, to provide a more comprehensive assessment of the independent effects of chemotherapy on HSM.

Discussion of Unmeasured Confounders:

In the discussion section, we have added a more detailed consideration of potential unmeasured confounders and their potential impact on our findings. We acknowledge the limitations inherent in observational studies and the potential for residual confounding despite our best efforts to control for known confounders.

We add again in the discussion section of the article the efforts we have made to ensure that the results are actionable. We have annotated in red font in the hope that our modifications will meet your requirements: "However, our study enhanced the PSM method to more accurately balance confounding factors between chemotherapy and non-chemotherapy groups. We also performed additional sensitivity analyses including different matching ratios (1:2 and 1:4) and IPTW using treatment weights to assess the robustness of our results under different assumptions. In addition, we performed stratified analyses based on key confounders to ensure that our findings were consistent across subgroups. Our CRR model now adjusts for a wider range of clinical and pathological factors, as well as treatment-related variables, to provide a more comprehensive assessment of the independent effects of chemotherapy on HSM. Therefore, our conclusion is highly feasible."

Finally, we ensured that all methods, analyses, and results related to the handling of confounding factors were transparently and thoroughly reported, in adherence with the STROBE (Strengthening the Reporting of Observational Studies in Epidemiology) guidelines, allowing for a clear understanding of our methodology and its potential limitations.

We believe that these revisions have significantly strengthened our manuscript by addressing the impact of confounding factors more comprehensively. We appreciate your guidance and the opportunity to improve the quality and robustness of our research.

---

## [Decision Letter · Decision Letter 3]

16 Jan 2025

Chemotherapy and Heart-Specific Mortality in Elderly Men with Prostate Cancer: A Propensity Score Matching Analysis

PONE-D-24-40438R3

Dear Dr. Zhanghuang,

We’re pleased to inform you that your manuscript has been judged scientifically suitable for publication and will be formally accepted for publication once it meets all outstanding technical requirements.

Kind regards,

Xing Xiong, M.D.

Academic Editor

PLOS ONE

Additional Editor Comments (optional):

Thanks for the authors' response to all the reviewers' concerns, and I am pleased to report that all the reviewers have now approved the publication of your paper. This paper can be accepted now.

Reviewers' comments:

Reviewer's Responses to Questions

**Comments to the Author**

1. If the authors have adequately addressed your comments raised in a previous round of review and you feel that this manuscript is now acceptable for publication, you may indicate that here to bypass the “Comments to the Author” section, enter your conflict of interest statement in the “Confidential to Editor” section, and submit your "Accept" recommendation.

Reviewer #1: All comments have been addressed

Reviewer #3: All comments have been addressed

2. Is the manuscript technically sound, and do the data support the conclusions?

Reviewer #1: Yes

Reviewer #3: Yes

3. Has the statistical analysis been performed appropriately and rigorously? 

Reviewer #1: Yes

Reviewer #3: Yes

4. Have the authors made all data underlying the findings in their manuscript fully available?

Reviewer #1: Yes

Reviewer #3: Yes

5. Is the manuscript presented in an intelligible fashion and written in standard English?

Reviewer #1: Yes

Reviewer #3: Yes

6. Review Comments to the Author

Reviewer #1: The authors have addressed all of my previous comments. I have no further comment for this manuscript.

Reviewer #3: The authors responded sincerely to my comments and recommendations, which I believe has strengthened the robustness of the study's findings.

I believe this is an important paper worthy of publication in our journal, PlosOne.

7. PLOS authors have the option to publish the peer review history of their article (what does this mean? ). If published, this will include your full peer review and any attached files.

**Do you want your identity to be public for this peer review?** For information about this choice, including consent withdrawal, please see our Privacy Policy .

Reviewer #1: No

Reviewer #3: No

---

## [Editor Report · Acceptance letter]

PONE-D-24-40438R3

PLOS ONE

Dear Dr. Zhanghuang,

I'm pleased to inform you that your manuscript has been deemed suitable for publication in PLOS ONE. Congratulations! Your manuscript is now being handed over to our production team.

Kind regards,

on behalf of

Dr. Xing Xiong

Academic Editor

PLOS ONE